# The Relationship between Alcohol Drinking Indicators and Self-Rated Mental Health (SRMH): Standardized European Alcohol Survey (SEAS)

**DOI:** 10.3390/healthcare10071260

**Published:** 2022-07-06

**Authors:** Danica Romac, Ljiljana Muslić, Diana Jovičić Burić, Mirjana Orban, Varja Đogaš, Sanja Musić Milanović

**Affiliations:** 1Teaching Institute of Public Health “Dr Andrija Štampar”, Mirogojska 16, 10000 Zagreb, Croatia; danica.romac@stampar.hr (D.R.); mirjana.orban@stampar.hr (M.O.); 2Croatian Institute of Public Health, Rockfellerova 7, 10000 Zagreb, Croatia; ljiljana.muslic@hzjz.hr (L.M.); diana.jovicic@hzjz.hr (D.J.B.); sanja.music@hzjz.hr (S.M.M.); 3School of Medicine, University of Split, Šoltanska 2, 21000 Split, Croatia; 4“Andrija Štampar” School of Public Health, School of Medicine, University of Zagreb, Rockfellerova 4, 10000 Zagreb, Croatia

**Keywords:** self-rated mental health, alcohol consumption, alcohol-related harms, preventive medicine, public health

## Abstract

Given that the self-perception of mental health is an important predictor of health outcomes and wellbeing, it is important to identify the indicators of mental health associated with alcohol consumption in order to reduce alcohol-related harms. This study used data from the cross-sectional RARHA SEAS survey (2015) in the Croatian general population, aged 18–64 years (n = 1500). Several aspects of drinking behaviors and alcohol-related harms were measured, as well as personal and sociodemographic factors. Logistic regression found a significant association between alcohol’s harm to others (AHTO) and poor self-rated mental health (SRMH) (OR = 0.752; 95% CI 0.601–0.941) in the total sample, as well as in the group of participants who rarely drank alcohol (OR = 0.504; 95% CI 0.322–0.787) in the last 12 months. More frequent consumers reported poor SRMH if they had at least one harmful effect from drinking (OR 0.538; 95% CI 0.295–0.980). Younger age, higher education, professional activity, and living with someone else in a household contributed to better SRMH. AHTO has been identified as a strong predictor of poor SRMH in the general population. Targeted public health and preventive measures are needed with specific approaches for different types of alcohol consumers.

## 1. Introduction

While alcohol is widely consumed, legally available, and socially accepted in many cultures, its effects on individuals and society are heavily debated. The physical, social, and emotional components of health may provide alcohol use and non-use with paradoxical meanings. Alcohol use may enhance social interactions, or it may provoke social regret and damage relationships [1].

Sufficient evidence supports the relationship of drinking patterns with alcohol-related consequences and self-reported health problems [2,3]. Previous studies have shown an association between problem drinking and various measures of poor wellbeing, including lower life satisfaction sleep problems, depression and anxiety, and other mental health problems [4,5,6]. Further evidence suggests that the link between mental health problems and alcohol consumption is very complex [5]. In addition, Saether et al. found that risky alcohol use was associated with slightly reduced life satisfaction and induced more mental health problems compared to low-risk alcohol use, while harmful alcohol use was associated with both low life satisfaction and more mental health problems [5,7]. Thus, considering the potentially harmful effects of alcohol consumption, especially when consumed in risky ways, it is important to monitor alcohol consumption behaviors [4,8].

Estimating the prevalence of heavy episodic drinking (HED), but also the volume of binge drinking and its share in overall alcohol consumption, provides us with information about risky drinking. Heavy episodic drinking, which consists of consuming, on one occasion, a volume of alcohol that is likely to lead to intoxication, is considered an initial step in developing alcohol use disorders (AUDs) but also to be risky from the perspective of public health [3,9] because the effects of this alcohol drinking pattern extend beyond the individuals and become a concern to the wider community [10].

The negative consequences of alcohol consumption not only affect the drinker, but may also exert negative effects on partners, family, friends, and the colleagues of drinkers. There is a wider range of harms experienced by someone other than the drinker [2], including physical, mental, emotional, and environmental types of harm. Studies report that people who have a heavy drinker in their lives experience reduced health and wellbeing [3,11,12].

Studying the effects of alcohol’s harm on others is of interest from a scientific perspective, as it is an important way in which the health and wellbeing of individuals may be affected, but also because it is likely to be important for informing Global Burden of Disease and Injury estimates and for public debate on alcohol policy [2].

General population health studies offer a tool to investigate the negative effects of alcohol on others and may assess alcohol-related harm using subjective health measures [11]. Self-rated mental health is a strong predictor of health outcomes and wellbeing [13], independent of the objective measures of health [14], and, validated in adults, is a useful indicator for monitoring the general mental health of the population [7,15,16].

Because many mental health conditions remain undiagnosed, self-rated assessment tools provide a useful and perhaps more revealing indicator of mental wellbeing, and they may capture a more comprehensive understanding of mental health in the general population, as well as help in identifying those individuals who are both in need of mental health resources and at risk of elevated alcohol harms. Catching these individuals early may enhance the relative efficacy of mental health treatment and potentially impact broader health expenditure [17]. Thus, the purpose of our research is to help reduce alcohol-related harm and to improve mental health and wellbeing by developing prevention programs for at-risk groups and promoting healthy lifestyles.

There are many studies that have investigated the correlation between patterns of alcohol consumption with mental health or mental wellbeing; this association with various measures is well established, but the form and nature of the association require further research [7]. The link between drinking behaviors and SRMH has not been evaluated enough, even though drinking patterns vary significantly from person to person. Furthermore, while diagnosed mental health conditions are often offered as a reason for nondrinking [18], the way people—especially those who choose to drink minimally or abstain and those who choose to drink more frequently—understand the interaction between alcohol drinking and their mental health and wellbeing mainly remains unclear and less well understood.

There is evidence to suggest that other dimensions of alcohol consumption, beyond the amount of alcohol consumed, may influence the risk of adverse health outcomes, hence the need for further research into the health risks associated with different patterns of alcohol consumption [19].

Alcohol consumption frequency is a widely used measure for monitoring alcohol consumption in the general population that may provide important, unique information about the risks of health outcomes. There is currently a lack of evidence to support recommendations about the frequency of alcohol consumption [19].

In most epidemiological literature, harmful drinking—a drinking pattern recognized as closely linked to alcohol-attributable diseases—is recorded using the measure ‘risky single occasion drinking’ (RSOD), which is based on drinking above a certain quantity. In contrast, subjective intoxication (SI), as an alternative measure, can provide additional information, including the drinker’s subjective perceptions and cultural influences on alcohol consumption and interindividual differences [20].

In addition, the importance of alcohol-related harm from others, which has been recognized in this study, has not yet been sufficiently explored, especially in Croatia, where alcohol plays a complex and important role in social life and culture.

However, none of the previous studies have comprehensively investigated the different dimensions of alcohol consumption patterns together, or their potential effects on the risk of adverse and subjective mental health outcomes, while controlling for other important personal and social determinants of mental health.

Identifying the nature, extent, and impact of alcohol-related harm is important for better understanding the direct and collateral damage alcohol places on society, and for managing and measuring the impact of alcohol policy and prevention activities. Given the differences in drinking patterns and the growing burden of alcohol-related disease, an assessment of the relationship between drinking habits and health consequences is needed to obtain information on treatment and prevention strategies [21].

The aim of this study is to examine the contribution of various indicators of alcohol consumption on the risk of poor SRMH outcomes among the general population by adjusting the analysis according to drinking frequency in order to examine how alcohol drinking patterns are associated with an increased risk of poor SRMH. Our study aims to develop an understanding of alcohol-related harms on SRMH by:Exploring the extent and nature of alcohol drinking behaviors and levels of harm related to SRMH in the context of social and personal factors;Examining the strength of the connection between alcohol drinking behaviors and harms with SRMH and the relationship with social and personal factors;Investigating the relationship between SRMH and different types of alcohol consumption frequency, controlling for socioeconomic and personal factors.

According to findings on the relationship between alcohol consumption and related harms that have been previously suggested [22], we have hypothesized that drinking behavior and alcohol-related harms predict subjective mental health. Based on previous studies, we anticipate that cross-sectional relationships between drinking behaviors and subjective mental health will be statistically significant, as well as the alcohol-related harms. Furthermore, we presume that social and personal factors may have an important role in subjective mental health.

## 2. Materials and Methods

### 2.1. Data and Samples

The data used in this study were collected during the Standardized European Alcohol Survey (SEAS) in 2015 as part of the EU Joint Action on Alcohol known as ‘Reducing Alcohol Related Harm’ (RARHA). The primary objective of RARHA was to obtain comparable baseline data for comparative assessment and monitoring of alcohol epidemiology across the European Union. Sampling procedure was country specific. This Croatian national survey was based on a two-stage, stratified random sample of the adult population, from 18 to 64 years of age. The sample size was set at 1500 individuals. First, regional strata were defined with four categories based on place of residence size for each stratum. Then, the required number of localities and random starting points within each locality, across all Croatian regions, were sampled. Finally, households were selected using the random walk method. Quota selection of respondents within a sampled household by age and gender was applied.

Household data collection was conducted via the CAPI method; interviews were carried out face to face, but computer-assisted from 26 May to 14 June 2015, using the standardized RARHA SEAS questionnaire [23], then translated and culturally adapted into the national language.

Some groups were excluded from sampling frames, such as the homeless or people living in an institution.

### 2.2. Measures

Self-rated mental health (SRMH) was measured using a single-item measure. Respondents were asked to rate their mental health on a five-point scale from very good, good, fair, poor, to very poor. For the present study, response categories were dichotomized as poor to fair SRMH (very poor, poor, fair) and good SRMH (good and very good) as in other similar studies [11].

To assess drinking behaviors over the past 12 months, several indicators were used from the RARHA main study questionnaire: frequency of drinking, heavy episodic drinking (HED), and alcohol intoxication (drunkenness).

The frequency of alcohol consumption offered data about usual overall consumption and about abstainers. Respondents were asked to indicate how often they drank any beverage containing alcohol, such as beer, wine, or spirits, even in small amounts, in the last 12 months. The frequency of alcohol use was analyzed with the categories: (1) daily or almost daily; (2) at least once a week (weekly); (3) at least once a month, but less often than once a week (monthly); (4) less than once a month (less frequently); and (5) never in the past 12 months (abstainers). Abstainers were not asked about drinking habits, and the other participants were assessed regarding drinking habits, such as HED and subjective intoxication. For the current study, frequency of alcohol consumption served as additional analysis, with two separate subsamples of different frequency drinking habits.

Heavy episodic drinking (HED) was assessed by reporting risky single occasion drinking (RSOD) in the past 12 months; participants were asked to indicate how many drinks they drank in the last 12 months. One drink was defined as 250 mL of beer, 100 mL of wine, or 30 mL of spirits. Heavy episodic drinking was considered when drinking 4 or more drinks in one occasion for women and 6 or more for men

Alcohol intoxication was assessed based on self-report of subjective drunkenness in the past 12 months, defined as experiencing feelings of unsteadiness or slurred speech after drinking. Answers were categorized into those who never drank alcohol, drank but never got drunk, and those who got drunk at least once in the past 12 months.

Individual harm to drinkers—the unwelcome consequences of drinking—was studied with a short, four-item screening instrument called ‘Rapid Alcohol Problems Screening Test’ (RAPS), which can identify problematic drinking. The RAPS scale consists of four simple questions dealing with unwelcome consequences of drinking in the past 12 months, with a total score ranging from 0 to 4, and including the following: feeling guilty, blacking out, failing to do what was normally expected, early morning alcohol consumption.

Alcohol’s harm to others (AHTO) was analyzed as: (a) exposure in the past 12 months to any known persons who drink a lot (heavy drinker) in the respondent’s life (including household members, family members outside the household, work- or schoolmates, and neighbors) and (b) being negatively affected by and experiencing any of the eight listed types of harms from others’ drinking: woken up at night because of someone else’s drinking; verbal abuse; physical harm; involvement in a serious argument; being a passenger with a driver who drank too much; being involved in a traffic accident; annoyed by vomiting, urinating, or littering; feeling unsafe in a public place, including public transportation. AHTO was estimated among respondents who reported they knew someone in the last 12 months who was a heavy drinker and that the drinking of that person had a negative influence on them. To measure exposure to heavy drinkers, respondents were categorized as those: (1) who did not know any heavy drinkers, (2) who knew a heavy drinker but answered that it did not have an influence on them, and (3) those who knew a heavy drinker and were negatively affected by that person.

Several personal and social variables were included in the study as covariates, based on previous literature [11]: age categorized as three groups (≤34 years, 35 to 49 years, and ≥50 years); level of education; professional activity; and number of household members. Level of education was categorized as low (completed primary school or lower), medium (completed secondary school), or high education (at least a short study at university or higher). Professional activity was assessed by asking respondents if they were professionally active or nonactive. Number of household members was measured based on how many persons they lived with. Those who lived alone were compared with those that had at last one household member.

For the current study, all categorical variables with three or more categories were adjusted for the logistic regression model by transforming them into quantitative ordinal representations. In that way, we included information about the order-level relationships between categories in our model. For example, the age variable with three age categories was transformed into an ordinal variable that included information about the natural ordering of ages, which means that the category of younger age came before the category of medium age. All dichotomized variables were used as original categorical variables in the regression model.

### 2.3. Statistical Analyses

Statistical analysis was carried out using SPSS Statistics ver.23.0 (ID: 729038; IBM Corporation, Chicago, IL, USA).

Descriptive and preliminary analyses with chi-square tests for association were performed to test the relationship between selected predictors and SRMH as an outcome variable. Three binary logistic regression analyses (stepwise backward) were performed separately from those predictors that were found to have a significant chi-square association with SRMH in the whole sample and in regard to frequency of alcohol drinking. OR and 95% CI were calculated for associations, and Nagelkerke R square for effect size. The results were analyzed at *p* < 0.05.

## 3. Results

### 3.1. Descriptive Analysis of the Sample

In the sample of 1500 respondents, 754 (50.3%) were male and 746 (49.7%) were female. The characteristics of the whole sample are shown in Table 1.

In terms of sociodemographic variables, the majority have medium education (44.9%), are professionally active (57.8%), and live with someone (93.2%). In addition, the majority had drinking habits and drank alcohol in the past 12 months in some way (78.6%), but they were less involved in heavy episodic drinking (24.2%) or alcohol intoxication (24.5%). Regardless of drinking frequency, some of them (21.4%) never drank during the past 12 months, 20.3% drank rarely, and 14% drank every day. Most of them drank weekly (26.4%). One or more unwelcome consequences of drinking in the past 12 months were reported by 16.4% of participants, and 25.4% were negatively affected by persons who drank too much. The self-rated mental health (SRMH) status was mostly rated as good (88.3%), especially among those who were younger, had a higher education, participated in professional activities, and lived in a household with someone.

Those who experienced HED (heavy episodic drinking) or drunkenness estimated lower SRMH than those who had never been intoxicated, which was as expected.

Those who drank weekly reported better SRMH than those who drank daily or rarely, possibly because of social and weekend activities, which are associated with better SRMH. Abstainers seemed to be a specific group according to their own characteristics (which has also been described previously in the literature [19]), so we did not engage in their more detailed analysis.

### 3.2. Test for Association with SRMH in Preliminary Analysis

To test which examined predictors statistically significantly correlated with SRMH, we conducted a chi-square test for association. The results are presented in Table 2. The greatest associations were found with the sociodemographic predictors of age (*p* < 0.01), educational level (*p* < 0.01), professional activity (*p* < 0.01), and household members (*p* < 0.01).

We also found statistically significant associations with SRMH, such as AHTO (*p* < 0.01), individual harms (*p* < 0.01), HED (*p* < 0.05), and drunkenness (*p* < 0.05). Due to a statistically significant association of SRMH with alcohol drinking frequency (*p* < 0.01) in the past 12 months, further analysis was performed in regard to different levels of alcohol drinking frequency. An additional analysis of relationships with SRMH was conducted for separate samples of respondents considering their level of alcohol drinking frequency: the group of rare (less frequent) drinkers and the group of more frequent (daily, weekly) drinkers.

In all analyses, age was found to be a consistently associated with SRMH, while other indicators—sociodemographic as well as alcohol indicators—differed depending on the sample (see Table 2).

### 3.3. Correlates of SRMH

Three binary logistic regression analyses were conducted. One was for the total sample and two for different samples, taking into account different levels of alcohol drinking frequency. Variables that were shown to be related to SRMH in the preliminary analysis (Table 2) were included as independent variables in the regression model. The final regression models for different samples are shown in Table 3.

Results show differences in SRMH predictors between different samples.

#### 3.3.1. SRMH in the Total Sample of Adults, 18–64 Years, in the General Population

The factors that increase and contribute to better SRMH are living with household members (OR 2.016, CI 1.190–3.416), participating in professional activities (OR 1.681, CI 1.182–2.391), and a higher level of education (OR 1.414, CI 1.095–1.827).

Being in a higher age group (OR 0.451, CI 0.350–0.581) and AHTO have a statistically significant negative correlation with SRMH in the total sample (OR 0.752, CI 0.601–0.941).

The classification percentage was 88.2%, and the Nagelkerke R square was 0.139.

#### 3.3.2. SRMH in the Sample of Adults Who Drank Alcohol Rarely (Less Frequently) in the Last 12 Months

Three factors included in the regression analysis remained in the final model. Professional activity increased SRMH (OR 3.167, CI 1.504–6.667), whereas the higher age group had a statistically significant negative impact on SRMH (OR 0.321, CI 0.193–0.535), as well as AHTO (OR 0.504, CI 0.322–0.787).

The classification percentage was 88%, and the Nagelkerke R square was 0.236.

#### 3.3.3. SRMH in the Sample of Adults Who Drank Alcohol More Frequently (Daily, Weekly)

In the final model, four out of five factors remained statistically significant. Living in a household with somebody increased SRMH (OR 5.255, CI 2.420- 11.280). A negative impact on SRMH was present in the higher age group (OR 0.518, CI 0.358–0.749) and in those experiencing individual harms from alcohol drinking (RAPS 1+) (OR 0.538, CI 0.295–0.980).

The classification percentage was 90.9%, and the Nagelkerke R square was 0.138.

We conducted an additional analysis regarding the frequency of drinking for two separate subsamples in which we compared regular alcohol drinkers (daily, weekly) to those who consumed alcohol rarely (less than monthly). The participants who responded “never” (Table 1) were grouped as abstainers from alcohol, and they were included in the total sample but excluded from these additional analyses (n = 320) because they were not in the focus of the research, and because of their specific characteristic, which has been shown in previous studies [ 4, 6, 24 Jani]. Considering that they abstained due to various reasons, they may have poorer wellbeing and a higher risk of negative outcomes due to reverse causality [4, RARHA]. The characteristics of abstainers are described in more detail in the RARHA Report synthesis [24].

Occasional drinkers, defined as participants who were reported as monthly alcohol consumers (e.g., one to three times a month) were also excluded from analysis because of their infrequent and inconsistent drinking habits. They are also known to underestimate their personal alcohol consumption based on previous literature [8]. The nature of our data prevented an accurate assessment of their weekly alcohol consumption, and the purpose of this additional analysis was to study the mental health risks associated with different drinking patterns among regular alcohol drinkers compared to those who drink less than monthly as the reference category.

Additional binary logistic regressions analysis, using subsamples with different drinking frequency habits, enabled us to make additional interpretations as to the nature of the connection between each indicator with respect to SRMH by analyzing differences in the predictions between subsamples.

A relationship between self-rated perceived poor mental health and alcohol drinking frequency was not found in the total sample, but the findings suggest a relationship between self-rated perceived poor mental health and individual harm when respondents belonged to the more frequent drinker group, as well as poor SRMH in those who drink less but have experienced alcohol harms from another heavy drinker in their life, and who had secondhand alcohol-related harms.

## 4. Discussion

The results from the present study support the existence of a significant negative association between alcohol-related harms and self-rated mental health (SRMH), but they also suggest a more complex relationship between drinking behaviors and SRMH. This is in line with some other studies, which showed that the relationship between alcohol drinking and SRMH may not be solely due to drinking behaviors, but also to the negative consequences of problematic drinking [4]. Furthermore, measuring the frequency of alcohol drinking provided important, unique information on the risks of alcohol consumption, and this has revealed differences in our predictions with respect to SRMH among two subsample groups, i.e., those who had rare (less frequent than monthly) and higher-frequency (daily, weekly) drinking habits.

In this study, individual harms were not found to be a significant predictor of SRMH in rare consumers of alcohol, which was expected because of their rare drinking. On the other hand, among more frequent drinkers, no significant impact on others’ drinking was found, which was unexpected. Other studies have also found that lighter and heavier drinkers self-manage their drinking in different ways [1].

The results also reveal that younger, highly educated, and professionally active participants, as well as individuals living with someone else in the household, had better SRMH in the total sample of the general population, which was expected and in line with other studies [11].

The drinking of others has been shown to be an important predictor of SRMH (in the total sample and the rarely drinking subsample), presenting a significant risk of lower SRMH among those who were exposed to and, at the same time, negatively affected by other heavy drinkers. The results in the final regression analysis suggest that the negative association of alcohol’s harm to others (AHTO) and SRMH persisted and remained significant among other significant predictors, indicating the strength of the connection between AHTO and poor SRMH (Table 3).

AHTO was identified as the strongest characteristic among the observed indicators of alcohol drinking in the general population, which showed the extent and nature of alcohol harms related to SRMH, not only on the individual level, but also on a broader level, i.e., on a social level. These results reinforce the findings in the literature that heavy drinking can have substantial effects on people who interact with drinkers, significantly affecting their quality of life and personal wellbeing [2,3,9,11,12].

Previous studies have shown associations between problem drinking, including heavy drinking, and various measures of poor wellbeing, lower life satisfaction, and mental health problems [5,6]. In this study, with regard to the alcohol indicators used in the regression models, the frequency of drinking behavior was not found to be a significant predictor of SRMH in interactions with other significant predictors in the model, nor with heavy episodic drinking or drunkenness. Although we found it to be significant when observed separately (Table 1 and Table 2), no significant predictive power was found in the final regression model involving other significant alcohol predictors with covariates such as age, education, professional activity, and living in a household with someone else. This could be explained by the fact that covariates, as stable and independent characteristics in the observed relationships, did not substantially change during interaction in their intercourse. In line with other studies, drinking behaviors may account for interindividual differences in habits, which probably depend on the circumstances and context in which they occur, thus changing during interaction [22]. Given that drinking behaviors were not shown to have significant importance for the prediction effect of SRMH, neither in total nor in additional subgroups, this should be further examined, as well as the role of methodology and predictor selection.

The mechanism through which our participants linked SRMH to other domains, such as education and career, suggest the importance of maintaining healthy lifestyles and personal relationships as protective factors in mental health [1]. These sociodemographic predictors may have encouraged a protective effect with respect to SRMH. Their relationship with drinking behaviors and alcohol harms requires further research in order to more accurately estimate the protective and risky factors of mental health.

To better understand the relationship between alcohol consumption indicators and SRMH, an analysis was also performed on subsamples with different alcohol drinking frequency habits. The results for those who drank rarely (less frequent than monthly) showed that the drinking of others had a higher impact on SRMH than personal drinking. They estimated their subjective mental health as poor if they were exposed to and negatively affected by heavy drinkers in their lives. This could be explained by not experiencing individual consequences that might affect SRMH because of their rare drinking, which was expected. It was also expected that predictors such as HED or drunkenness were not significant in this group due to their drinking habits.

Contrary to rare drinkers, those who drank alcohol more frequently (daily and weekly) rated their SRMH as poor if they had at least one unwelcome consequence of personal drinking. Thus, in more frequent alcohol consumers, risk assessments for lower SRMH indicate that such the association is due to individual drinking harms, which is in line with previous studies [5,12].

The different effects on SRMH between these two groups may be explained not only by the different frequency of drinking, but also by the different and complex underlying mechanisms that depend on individual and environmental factors, as well as associations with specific life circumstances [3,8,21].

Interestingly, in our study, more frequent drinkers did not estimate alcohol’s harm to others (AHTO) as a risk for poor SRMH, which is not in line with previous studies that found an association between frequent drinking and higher odds of experiencing AHTO [12]. A possible explanation for these differences may be the seemingly different perspectives between participants with opposing drinking habits, possibly due to interactional occurrences on the level of individuals and their relationships, but also possibly due to a variety of social factors, weekend activities, drinking contexts, and specific types of harm that we have not extensively researched in the present study. The context in which AHTO is experienced, including harm type, source and length of exposure, and its relationship to both immediate and long-term health and wellbeing, requires further consideration and research [21]. This also highlights the issue of being careful when comparing results from different studies because of different methodologies, survey contexts, and sources of bias [11].

Some studies suggest that drinking frequency may be a proxy variable for other social factors, and that there is also a possible intermediate factor between drinking frequency and SRMH [8,25]. In the present study, the regression model showed that participants with different drinking frequency habits had differences in the sociodemographic characteristics of their SRMH. Participation in professional activities significantly promoted better SRMH in those who rarely drink (or who perhaps drink less frequently due to lack of time). The type of personal relationship, such as living with someone else in a household, was associated with higher odds of reporting better SRMH among those who drank regularly, which suggests that living with somebody may be a protective factor for those who drink more frequently; living alone and being socially disconnected may be linked to increased solitary drinking, which has been described in the literature [26]. Younger age proved to be a significant and constant predictor of better SRMH in all groups (total sample, less and more frequent drinkers), thus confirming the data showing that self-rated good mental health decreases with age [14] regardless of differences in alcohol consumption frequency habits.

This study offers new information about the complex relationship between SRMH and different drinking indicators while also controlling for other important factors related to SRMH. The finding that AHTO has the strongest contribution to poor SRMH could help us to understand the real magnitude of the harm that heavy and problematic alcohol consumption has on individuals and society.

However, in reality, the relationship between alcohol consumption and mental health is very complex and undeniably entangled with broader changes in social practices and norms, not only with respect to the factors used in this study, but also to various other factors. One of the complexities of alcohol drinking habits is their potential benefit to social wellbeing [27], and some studies have found that certain individuals respond to a decline in life satisfaction by drinking alcohol in an attempt to improve their lives [5]. Many studies have indicated that increased levels of mental health problems are associated with a heightened risk of alcohol-related problems, and the association may be a result of self-medication. Mental health problems and alcohol misuse may have a negative reciprocal effect on each other through the creation of a negative cycle [5].

While some consider alcohol socially pleasurable and a facilitator of fun, which is important to their wellbeing and their social lifestyles [1], research from Denmark and Sweden also highlight the importance of physical, mental, and social health, as well as wellbeing, among abstainers and light drinking choices [18].

This study contributes to the understanding of the relationship between positive predictors, potentially risky behaviors, and alcohol-related harms on SRMH. Our results have the potential to inform tailored preventative approaches that address groups at risk for poor SRMH. In addition to their scientific value and practical significance, these studied domains represent major public health concerns.

Studying individuals with poor and good SRMH could serve as a supportive strategy to achieve favorable healthy behaviors, as well as to provide information on the risks associated with alcohol consumption, thus correcting misconceptions that may contribute to risky lifestyles. Prevention strategies could also raise awareness of mental health issues and responsible alcohol consumption, which could serve as a motivator to change problematic drinking behaviors in order to maintain and improve people’s subjective mental health and wellbeing.

Programs and policies targeting both alcohol use and mental health should consider how to reduce both the prevalence and impact of alcohol-related harms in the general population [28]. Although SRMH is a useful tool for assessing the mental health needs of the population, poor SRMH may not have a universal meaning; thus, we need further research to investigate correlations and to clarify the causal link between alcohol consumption and indicators of poor SRMH.

Our study has several limitations. First, our data are cross-sectional, which limits our ability to make causal inferences. Furthermore, we cannot discuss an individual’s cumulative mental health history and can only discuss drinking behaviors within the past 12 months.

We additionally acknowledge that a subjective measure of mental health does not inform us of the specific mental health conditions respondents were experiencing, so the problem as related to reverse causation could not be discussed in detail to provide a more balanced interpretation of the result [29].

Further limitations in this study are that all the information was self-reported and the indicators used may not have covered all the domains. Another shortcoming is the list-wise approach to missing data, which has been described in the literature [30].

Finally, mental health assessed using these instruments may be more or less sensitive to varying aspects of a given condition. Moreover, we had to rely on the respondents’ self-assessments, which could be biased.

In general, research on subjective mental health has been conducted mostly in Western nations, and studies on ethnic differences are still rare, so caution should be applied when comparing our results with other cultures [22].

The study has several strengths. A key strength of this study was the utilization of a large representative sample of the entire Croatian general population, with a relatively high response rate, which aided the external validity of our study. To the best of our knowledge, this is the first study to include a composite measure for both SRMH and drinking behaviors with alcohol harms, covering SRMH more broadly rather than focusing only on single aspects. This will help inform policy makers and healthcare workers about general trends in the population.

## 5. Conclusions

In this study, we found a strong association between alcohol’s harm to others (AHTO) and self-rated mental health (SRMH); those who were negatively affected by heavy drinkers were more likely to report poor SRMH compared to individuals who did not experience harms from a heavy drinker. Additionally, we found that the experience of harm attributable to drinking differs whether we examine harms among rare drinkers or more frequent drinkers.

Thus, the likelihood of reporting poor SRMH and AHTO increased in less frequent drinkers, who are at risk of poor SRMH because of another’s heavy drinking. Those who drink more frequently appear to be at risk of poor SRMH if they have at least one individual harm due to their own drinking. The results also demonstrate that being younger, having higher education, being professionally active, and living with someone else in the household contribute to better subjective mental health. In general, findings of different forms of associations indicate that alcohol use is experienced in various ways but can be risky for SRMH, not only among frequent consumers, but also among those who drink rarely if they are exposed to and negatively affected by heavy drinkers.

However, further research is needed to investigate the dynamics of the determined relationships between alcohol consumption and self-rated mental health, focusing not only on those who drink more frequently, but also on those who are close to heavy drinkers in order to support good mental health in the long run.

## Figures and Tables

**Table 1 healthcare-10-01260-t001:** Characteristics of the sample with SRMH and categories according to drinking status and personal and sociodemographic variables.

	Variables	Category	n (%) ^1^	SRMH (n%)	
				Poor to Fair	Good
Personal variables	Gender	Male	754 (50.3)	85 (48.3)	669 (50.7)
		Female	746 (49.7)	91 (51.7)	654 (49.4)
	Age group	Younger (18–34)	534 (35.6)	22 (12.5)	512 (38.7)
		Medium (35–49)	471 (31.4)	44 (25.0)	427 (32.3)
		Older (50+)	495 (33.0)	110 (62.5)	384 (29.0)
Sociodemographic variables	Education level	Low	561 (37.4)	98 (55.7)	463 (35.1)
		Medium	672 (44.8)	56 (31.8)	615 (46.6)
		High	264 (17.6)	22 (12.5)	242 (18.3)
	Professional activity	Nonactive	629 (41.9)	106 (60.6)	522 (39.7)
		Active	863 (57.5)	69 (39.4)	794 (60.3)
	Household members	Living alone	102 (6.8)	24 (13.7)	78 (5.9)
		Living with someone	1394 (92.9)	151 (86.3)	1242 (94.1)
Exposure to and negatively affected by another heavy drinker (HD)	Alcohol’s harm to others (AHTO)	Does not know any HDs	667 (44.5)	65 (36.9)	602 (45.8)
		Knows another HD without influence	445 (29.7)	46 (26.1)	399 (30.3)
		Knows another HD with a negative influence	379 (25.3)	65 (36.9)	313 (23.8)
Personal drinking consequences	Individual harms for a drinker (RAPS)	Abstainers	320 (25.6)	52 (36.1)	267 (24.2)
		None (RAPS—all negative)	723 (57.9)	63 (43.8)	660 (59.8)
		At least one (RAPS 1+)	205 (16.4)	29 (20.1)	176 (16.0)
Alcohol drinking behaviors	Frequency of alcohol drinking in the past 12 months	Never	320 (21.4)	52 (29.7)	267 (20.2)
		Rare	304 (20.3)	39 (22.3)	265 (20.1)
		Monthly	268 (17.9)	20 (11.4)	248 (18.8)
		Weekly	395 (26.4)	35 (20.0)	360 (27.3)
		Daily	210 (14.0)	29 (16.6)	181 (13.7)
	Heavy episodic drinking in the last 12 months (HED)	Abstainers	320 (21.5)	52 (30.4)	267 (20.9)
		Never drink risky	777 (53.7)	85 (49.7)	692 (54.2)
		1+ time(s)	351 (24.2)	34 (19.9)	317 (24.8)
	Alcohol intoxication in the last 12 months (drunkenness)	Abstainers	320 (21.5)	52 (29.5)	267 (20.4)
		Never intoxicated	801 (53.8)	84 (47.7)	717 (54.6)
		1+ time(s)	368 (24.7)	40 (22.7)	328 (25.0)
Subjective mental health	Self-rated mental health (SRMH)	Poor to fair	176 (11.7)	176 (11.7)	1323 (88.3)
		Good	1323 (88.3)		

^1^ In regard to the total number of respondents to a specific question.

**Table 2 healthcare-10-01260-t002:** Association between personal, sociodemographic, and other variables regarding alcohol and SRMH in the whole sample of adults, aged 18–64 years, in the subsample of those who rarely drank alcohol (less frequent than monthly) and those who drank alcohol more frequently (daily, weekly) in the last 12 months.

	SRMH Total Sample			SRMH Less Frequent Drinking Subsample (Rarely)		SRMH More Frequent Drinking Subsample (Daily, Weekly)	
	χ^2^	*p*		χ^2^	*p*	χ^2^	*p*
Gender	0.321	0.571	Gender	0.673	0.412	1.943	0.163
Age group	85.367 **	<0.01	Age group	22.737 **	<0.01	18.824 **	<0.01
Education level	28.134 **	<0.01	Education level	4.683	0.096	12.522 **	0.002
Professional activity	27.691 **	<0.01	Professional activity	10.978 **	0.001	7.985 **	0.005
Household members	14.807 **	<0.01	Household members	1.226	0.268	24.379 **	<0.01
Alcohol’s harm to others (AHTO)	14.183 **	0.001	Alcohol’s harm to others (AHTO)	9.855 **	0.007	3.373	0.185
Heavy episodic drinking (HED)	8.257 *	0.016	Heavy episodic drinking (HED)	0.244	0.621	0.114	0.735
Alcohol intoxication (drunkenness)	7.825 *	0.020	Alcohol intoxication (drunkenness)	0.699	0.403	0.489	0.484
Individual harms (RAPS 1+)	14.096 **	0.001	Individual harms (RAPS 1+)	2.092	0.148	6.046 *	0.014
Frequency of alcohol drinking in the past 12 months	15.558 **	0.004					

* *p* < 0.05; ** *p* < 0.01.

**Table 3 healthcare-10-01260-t003:** Odds ratios for better SRMH in adults aged 18–64 (total sample) and in two subsamples of those who drank alcohol rarely (less frequent than monthly) and in those who drank alcohol more frequent (daily and weekly), in the last 12 months.

SRMH (Self-Rated Mental Health)								
Total Sample			Less Frequent Drinking Subsample (Rarely)			More Frequent Drinking Subsample (Daily, Weekly)		
	OR (95% CI)	*p*		OR (95% CI)	*p*		OR (95% CI)	*p*
Age group	0.451 ** (0.350–0.581)	<0.01	Age group	0.321 ** (0.193–0.535)	<0.01	Age group	0.518 ** (0.358–0.749)	<0.01
Education level	1.418 * (1.074–1.871)	0.014				Education level	1.543 * (1.005–2.367)	0.047
Professional activity	1.531 * (1.040–2.254)	0.031	Professional activity	3.167 ** (1.504–6.667)	0.002			
Household members	2.385 ** (1.328–4.211)	0.003				Household members	5.255 ** (2.420–11.280)	<0.01
Alcohol’s harm to others (AHTO)	0.752 * (0.601–0.941)	0.013	Alcohol’s harm to others (AHTO)	0.504 ** (0.322–0.787)	0.003	Individual harms (RAPS 1+)	0.538 * (0.295–0.980)	0.043

* *p* < 0.05; ** *p* < 0.01; never or no was the reference category; confidence interval values are shown in brackets.

## Data Availability

The Croatian RARHA SEAS data can be made available upon request. Please contact the Croatian Institute of Public Health directly for the data.

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
