# Peer review of "The Relationship between Alcohol Drinking Indicators and Self-Rated Mental Health (SRMH): Standardized European Alcohol Survey (SEAS)"

_healthcare, 2022, doi:10.3390/healthcare10071260_

Round 1

Reviewer 1 Report

The topic of the study is very important and the results are very interesting. There is still some space to improve description and to consider some modification of analyzes. My suggestions would as follows:

1) The aim of the study (line 11-116) – in fact only relationships between alcohol related factors and socio-demographic  factors and SRMH as target variable are examined. There are not any interactions between factors included into models.

2) Initial categories of factors and their recoding for bivariate analyses and logistic regression models could be more clearly described. E.g.  looking at descriptions of frequency of drinking (line 150-153), HED (line 154-158) or alcohol intoxication (line 159-162) we don’t know whether abstainer are included or excluded. Additionally in the dichotomization of frequency of drinking (line 150-153) category once a month is omitted (those who consumed alcohol less frequently (less than once a month) and those who consumed alcohol more frequently drinking (weekly, daily or almost every day); definition of RAPS scale (line 165-168) – The value of cut-off point is not mentioned; definition of last category of AHTO – (line 181-182) there is not clear if only one known heavy drinkers who affected respondents is enough to be included. (“… were negatively affected by them” suggested that more is needed).   

3) Table 1. – percentages in column SRMH should be calculated in another way, that means in direction of factor instead of target variables. In other words  we need to know what is % of SRMH among male and females or among particular age groups. So the percentages should sum-up to 100% in rows.

4) Table 2. – the question is if the data in this table are categorized in the same way as in table 1. There is not percentage distributions in table 2.

5) Table 3. Data in the table and description under table suggests that factor variables were dichotomized (never or no was reference category). But age group don’t fit to this interpretation – if it was  dichotomized,  where was the cut-off point and which category was the reference category). Incidentally, why dichotomization for factor variables was used, if one was used. E.g. 3 categories of age could give interesting results.  Sometimes the old people are similar to young people at differ  from those who are between.  It is not clearly mentioned that “Less frequent drinking subsample” contains abstainers.

6) Several times in the text we meet suggestions that influence of factor variables were explores in mutual interaction (e.g. line 315-321, line 347-349). Logistic regression don’t examine relationships between factor variables or common impact on target variables until interaction terms are not included into model. On the basis of the models presented in article we can say only what is impact of any factor variable, when impact of other variables is under control.  

Reviewer 2 Report

This is an interesting study examining the association between alcohol drinking indicators and self-rated health. I have several major concerns that require a major revision:

1. In the introduction, the authors mentioned that other dimensions of alcohol consumption, beyond the amount of alcohol consumed are important. However, there is little discussion on what are the indicators and why they are important in the introduction. This part should be elaborated.

2. There is a need to mention how missing data was treated in the current study. If listwise deletion was used for missing data, it will be useful to report the percentage of data that was removed. A  brief justification on the decision to use listwise deletion will be helpful too. Relevant paper:
Newman, D. A. (2014). Missing data: Five practical guidelines. Organizational Research Methods, 17(4), 372-411.

3. It will be useful for the authors to report zero-order correlations of all main variables in the study.

4. I have serious concern on why self-rated health was dichotomized. As dichotomization of continuous variable has been shown to inflate type 1 error, it will be important for the authors to provide more justification on why this is necessary. Relevant paper:
Royston, P., Altman, D. G., & Sauerbrei, W. (2006). Dichotomizing continuous predictors in multiple regression: a bad idea. Statistics in Medicine, 25(1), 127-141

5. Another important point to discuss in the Discussion section is the possibility of reverse causation which is a common limitation in the literature. While the authors suggested that alcohol consumption may affect self-rated mental health it is equally plausible that self-rated mental health is the antecedent of alcohol consumption. This problem related to reverse causation should be discussed in details to provide a more balanced interpretation of the result. This is a critical point to discuss. Please see the following paper for a relevant discussion related to reverse causation: Quek, F. Y., Tng, G. Y., & Yong, J. C. (2021). Does social media use increase depressive symptoms? A reverse causation perspective. Frontiers in Psychiatry, 12, 335.

6. There is a need for the authors to justify the use of stepwise backward regression. Also, it will be important to justify the choice of covariates. For example, household income and health-related variable were not controlled in the current study,

7. The authors should also clarify the inclusion and exclusion criteria of the sampling. The technique used for random sampling should be clarified too.

Round 2

Reviewer 2 Report

The authors have adequately addressed all my comments. I appreciate all their efforts. The manuscript is ready for publication.